# A mediation analysis evaluating change in self-stigma on diabetes outcomes among people with depression in urban India: A secondary analysis from the INDEPENDENT trial of the collaborative care model

Scott Halliday[1]*, Deepa Rao[1,2], Orvalho Augusto[1,3,4], Subramani Poongothai[5,6], Aravind Sosale[7], Gumpeny R. Sridhar[8], Nikhil Tandon[9], Rajesh Sagar[10], Shivani A. Patel[11], K. M. Venkat Narayan[11], Leslie C. M. Johnson[11,12], Bradley H. Wagenaar[1,13], David Huh[14], Brian P. Flaherty[15], Lydia A. Chwastiak[1,2], Mohammed K. Ali[11,12], Viswanathan Mohan[5,6], INDEPENDENT Study Group

**1** Department of Global Health, University of Washington, Seattle, Washington, United States of America, **2** Department of Psychiatry and Behavioral Sciences, University of Washington, Seattle, Washington, United States of America, **3** Faculdade de Medicina, Universidade Eduardo Mondlane, Maputo, Mozambique, **4** Manhiça Health Research Centre (CISM), Maputo, Mozambique, **5** Madras Diabetes Research Foundation, Chennai, India, **6** Dr. Mohan's Diabetes Specialities Centre, Chennai, India, **7** Diabetes Care and Research Center, Diacon Hospital, Bangalore, India, **8** Endocrine and Diabetes Center, Visakhapatnam, India, **9** Department of Endocrinology and Metabolism, All India Institute of Medical Sciences, New Delhi, India, **10** Department of Psychiatry, All India Institute of Medical Sciences, New Delhi, India, **11** Emory Global Diabetes Research Center, Woodruff Health Sciences Center and Emory University, Atlanta, Georgia, United States of America, **12** Department of Family and Preventive Medicine, Emory University, Atlanta, Georgia, United States of America, **13** Department of Epidemiology, University of Washington, Seattle, Washington, United States of America, **14** School of Social Work, University of Washington, Seattle, Washington, United States of America, **15** Department of Psychology, University of Washington, Seattle, Washington, United States of America

* shallid@uw.edu

**Data Availability Statement:** Deidentified participant data are available upon email request to

## Abstract

Self-stigma–the internalization of negative community attitudes and beliefs about a disease or condition–represents an important barrier to improving patient care outcomes for people living with common mental disorders and diabetes. Integrated behavioral healthcare interventions are recognized as evidence-based approaches to improve access to behavioral healthcare and for improving patient outcomes, including for those with comorbid diabetes, yet their impact on addressing self-stigma remains unclear. Using secondary data from the Integrating Depression and Diabetes Treatment (INDEPENDENT) study–a trial that aimed to improve diabetes outcomes for people with undertreated and comorbid depression in four urban Indian cities via the Collaborative Care Model–we longitudinally analyzed self-stigma scores and evaluated whether change in total self-stigma scores on diabetes outcomes is mediated by depressive symptom severity. Self-stigma scores did not differ longitudinally comparing Collaborative Care Model participants to enhanced standard-of-care participants (mean monthly rate of change in Self-Stigma Scale for Chronic Illness-4 Item scores; B = 0.0087; 95% CI: -0.0018, 0.019, $P$ = .10). Decreases in total self-stigma scores over 12

the Emory Global Diabetes Research Center: egdrc@emory.edu. Requesters should be employees of a recognized academic institution or health service organizations. Requesters must have experience in medical research and must be able to demonstrate their ability to carry out the proposed use of the requested dataset through their peer-review publications in the area of interest. Data sharing will only be for the purposes of health and medical research and within the constraints of the consent under which the data were originally gathered. The requester(s) will be required to enter into a Data Sharing Agreement with Emory University. The statistical software code used in this analysis is contained in Supplemental File 2.

**Funding:** This study was funded by the National Institute of Mental Health (grant R01MH100390) with LC, MA, and VM as Principal Investigators receiving the grant and the University of Washington Behavioral Research Center for HIV (BIRCH), a National Institute of Mental Health-funded program (P30 MH123248). The funders had no role in study design, data collection and analysis, decision to publish, or preparation of the manuscript. The content is solely the responsibility of the authors and does not necessarily represent the official views of the National Institutes of Health.

**Competing interests:** SH is a technical adviser for the non-profit organization Possible for which he receives no compensation. The authors have declared that no competing interests exist.

months predicted diabetes outcomes at 12 months (HbA1c, total effect; B = 0.070 95%CI: 0.0032, 0.14; $P < .05$), however depressive symptoms did not mediate this relationship (average direct effect; B = 0.064; 95% CI: -0.0043, 0.13, $P = .069$). Considering the local and plural notions of stigma in India, further research is needed on culturally grounded approaches to measure and address stigma in India, and on the role of integrated care delivery models alongside multi-level stigma reduction interventions.

**Trial registration :** ClinicalTrials.gov, NCT02022111. https://clinicaltrials.gov/study/NCT02022111.

## Introduction

Negative attitudes and beliefs held by a community or group about a disease or condition that is devalued may lead to stigmatization, but the process of internalizing those attitudes and beliefs by a person living with that disease may result in self-stigma. Self-stigma can be experienced by people with various conditions including depression, anxiety [1–3], and diabetes [4]. While self-stigma may limit access to care and treatment-seeking, it can also negatively impact ongoing participation for those already receiving treatment and their care-related outcomes [3,5,6] Importantly, negative attitudes and beliefs associated with self-stigma can be reinforced by healthcare providers. For example, providers may treat diabetic patients unfairly due to their weight, which may further the internalization of blame and judgement [7]. Healthcare systems may be under-resourced to treat mental disorders, which can re-affirm beliefs among patients with mental disorders that they are not worthy of treatment [8,9].

In India, where estimates indicate that 10% of Indian adults live with a common mental disorder [10], 11% of Indian adults live with diabetes [11], and many live with both comorbid depression and diabetes [12], studies suggest that stigma might be prevalent among people living with common mental disorders [2,13–17] and diabetes [18–21]. Yet Indian primary care providers may receive little training in diagnosing or managing mental disorders, face shortages of psychotropic medications, deliver care in settings with little privacy for patients, and reinforce misconceptions about mental disorders or hold discriminatory attitudes towards people with mental disorders [22–25]. As a result, visiting a healthcare provider in India can lead to or exacerbate the stigmatization of common mental disorders [22–25]. Stigma thus represents an important barrier, among other access-related barriers (e.g., an insufficient number of mental health and diabetes specialists given the burden of disease) to improving the quality of patient outcomes for people living with these conditions [10,26].

Integrated behavioral healthcare (BH) interventions are healthcare system interventions where specialty BH providers are incorporated into primary care or medical specialty settings with the goal of improving care coordination, represent options to improve patient outcomes [27] and are widely endorsed by national and international bodies [5]. There is limited evidence suggesting that integrated interventions can reduce stigma around accessing mental healthcare services, including in South Asia [28–30]. However, the effect of integrated BH interventions on reducing self-stigma specifically remains understudied [31–33]. The Integrating Depression and Diabetes Treatment (INDEPENDENT) study demonstrated the effectiveness of a specific integrated care model, the Collaborative Care Model (CoCM), in improving clinical outcomes for patients living with comorbid depression and type 2 diabetes who received treatment in four tertiary urban diabetes clinics in India [34]. It remains unclear

whether or not integrated BH interventions like CoCM lead to improvements in self-stigma over time and if changes in self-stigma predict these patient health outcomes.

In the INDEPENDENT trial, four urban diabetes clinics in India tested CoCM as an intervention in part as undertreated depression was viewed as a barrier to improving clinical outcomes for patients with type 2 diabetes [34]. Considering that healthcare providers and facilities in India can be the cause of or perpetuate stigma, particularly among people living with common mental disorders [22,23,25], we proposed that an integrated BH intervention such as CoCM may reduce levels of self-stigma due to patients receiving depression treatment in a familiar clinical setting rather than at a potentially stigmatizing specialist provider via referral. As such, depression treatment would mediate the relationship between change in self-stigma and diabetes outcomes.

We drew on data from the INDEPENDENT trial implementing CoCM at four urban diabetes clinics in India to address two objectives. First, we analyzed self-stigma scores longitudinally to compare participants assigned to CoCM versus enhanced standard-of-care. Second, we evaluated whether change in self-stigma on type 2 diabetes outcomes is mediated by depressive symptoms.

## Methods

### Data sources and participants

We used secondary data collected during the INDEPENDENT trial (ClinicalTrials.gov Identifier: NCT02022111) for integrating care for depression among adult participants with type 2 diabetes via CoCM. The full study protocol [35] and primary trial results [34] have been previously published elsewhere. In this pragmatic trial, patients at four urban diabetes clinic sites (one public hospital in Delhi and three private clinics in Chennai, Visakhapatnam, and Bengaluru) were individually randomized to receive either 12 months of enhanced standard care or CoCM and then passively followed up to 36 months. The enhancement to standard care involved notifying diabetes physicians of their patients' depressive symptoms at enrollment; delivering continuing medical education on depressive symptom recognition and treatment; and providing either care for depressive symptoms or referral to a specialist clinic. To be eligible, participants must have been 35 years of age at the time of enrollment (recruitment started on 9 March 2015 and ended on 31 May 2016), had a confirmed type 2 diabetes diagnosis, and had at least one poorly controlled cardiometabolic parameter out of the following: hemoglobin A1c (HbA1c) $\geq$8%, systolic blood pressure (SBP) $\geq$ 140 mmHg, or low-density lipoprotein (LDL) cholesterol $\geq$ 130 mg/dl as documented in a medical chart review. Eligible participants were screened for depression using the Patient Health Questionniare-9 (PHQ-9) and those with scores of 10 or above, indicating moderate to severe depression, were invited to participate. Participants were excluded based on the following criteria from the chart review: currently receiving treatment for depression; alcohol or substance use disorders; cognitive disorder; bipolar, schizophrenia spectrum, or psychotic disorders; kidney failure; or cardiovascular disease including any myocardial infarction, unstable angina, or stroke within the past 12 months. There were three protocol deviations that occurred after screening but prior to randomization (participants were excluded): one participant was less than 35 years of age and two participants were on antipsychotic medications. The final study-related data collection at 24 months was completed in July 2018.

### Ethics

The institutional ethics committees at each of the participating sites and the coordinating centers (Madras Diabetes Research Foundation and Emory University) approved the study. We

received written informed consent from all eligible participants prior to enrollment in the study. Additional information regarding the ethical, cultural, and scientific considerations specific to inclusivity in global research is included in the (S1 Checklist).

## Measures

We assessed self-stigma using the 4-item Stigma Scale for Chronic Illness (SSCI-4; see S1 File), a brief, non-illness specific instrument resulting from iterations over the previously developed 24-item and 8-item stigma scales [1,36]. The SSCI-4 has not been routinely used in India nor are there additional psychometric studies published of the shortened scale, but it was chosen for brevity and as researchers had completed a non-Western cultural adaptation in a previous study [37]. The SSCI-4 assesses four sub-constructs: social avoidance, blame, feeling left out, and embarrassment. The SSCI-4 prompts participants to self-report stigma using a 5-item Likert scale rating. These scores are then summed for an overall self-stigma score on a scale from 4 (no self-stigma) to 20, with higher scores indicating higher levels of self-stigma. The scale also includes a prompt for participants to consider their most stigmatizing condition while completing items (depression or diabetes). The 4-item scale at baseline (Cronbach's alpha = .841) demonstrated good internal consistency.

Depressive symptom severity was assessed using the PHQ-9 while HbA1c levels were measured via blood draw and laboratory method. The PHQ-9 was used both for screening and for measuring treatment response, given the US National Committee for Quality Assurance's guidelines defining depression treatment response as a $\geq$ 50% reduction in PHQ-9 score at follow-up and remission as a follow-up PHQ-9 score of $\leq$ 5 [38,39]. The PHQ-9 has previously been validated for screening depression in Chennai, India and demonstrated invariance to screening for depression among those diagnosed with or at risk for diabetes in Kerala, India [40,41]. Multiple demographic variables were collected at baseline including age, sex, monthly household income in Indian Rupees (INRs), and level of educational attainment. Whereas the PHQ-9 and HbA1c were measured at each study measurement point (baseline, six months, 12 months, 18 months, 24 months, and 36 months), self-stigma scores were measured yearly (baseline, 12 months, 14 months, and 36 months).

## Data analysis

We conducted a longitudinal analysis comparing self-stigma over time by intervention assignment. We used a linear mixed effects model, which included a random intercept, to account for correlation within individuals over time. Although these data originated from a randomized trial, we adjusted for the following covariates in a sensitivity analysis: age, sex, monthly household income, and level of educational attainment [42].

We propose that an important part of why CoCM was effective at improving diabetes and depression outcomes was because CoCM led to improvements in self-stigma, and this change in turn predicted the patient clinical outcomes of interest. Our mediation analysis—which was guided by Baron and Kenny's mediation testing approach—sequentially evaluated the following: 1) the effect of the predictor of interest, change in self-stigma (SSCI-4 scores) over 12 months, on the outcome of diabetes (HbA1c levels) at 12 months (total effect); 2) the effect of change in self-stigma on the mediator, depressive symptoms (PHQ-9) at 12 months; and 3) the effect of change in self-stigma on diabetes accounting for depressive symptoms (direct effect) as shown in Fig 1 below [43]. The average causal mediation effects (or indirect effect) is the total effect minus the direct effect [43]. Given the primary trial design and to account for variation in depressive symptom severity and diabetes scores at baseline, we used an ANCOVA approach to adjust for these baseline values by adding them as fixed effects accordingly instead

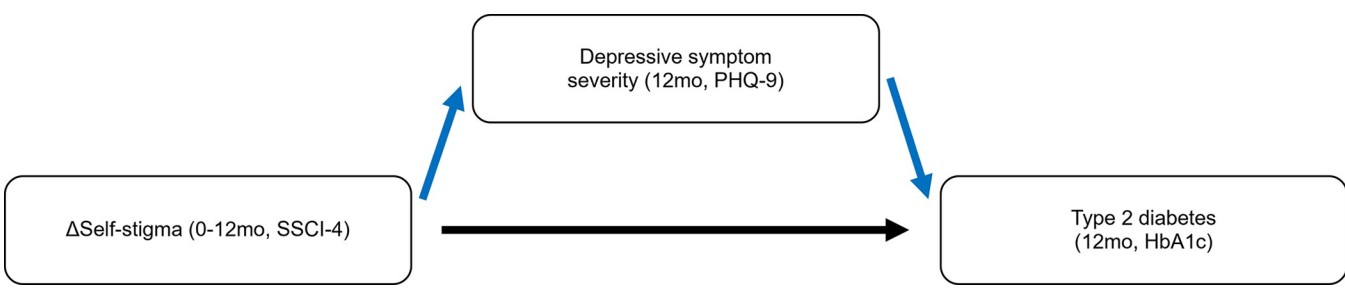

**Fig 1. Path model diagram of proposed mediation analysis.**

of using change scores for both which conserved power and improved statistical precision [44]. To determine the statistical significance and generate robust confidence intervals, we used a bootstrapping method with 1000 resampled data sets as described by Preacher and Hayes [45].

For both the mediation and longitudinal analyses, we analyzed the data for patterns of missingness and found that over 6% of individuals had missing SSCI-4 scores in a monotone pattern consistent with drop-out. As a result we assumed the data were missing at random (MAR), indicating that the covariates are good predictors of missingness, and chose to multiply impute the missing data given the availability of covariates in the data [46–48]. We used an expectation-maximization bootstrap algorithm based on a multivariate normal distribution to generate 30 imputed datasets for which the analyses were run followed by pooling of results [49]. In addition to SSCI-4 scores, the following variables were also multiply imputed due to missingness: stigmatized condition; level of educational attainment; and HbA1c levels and PHQ-9 scores at 12, 24, and 36 months. We performed all analyses using R version 4.2.2 [50] and our annotated code, which includes a list of all packages used, is available in S2 File.

## Results

Four hundred and three participants consented to the trial and were randomized to receive either CoCM or enhanced standard-of-care. Participants across the CoCM and enhanced standard-of-care groups were comparable at baseline as participants on average as shown in Table 1: were female (59%), had completed secondary school (51%), had a mean monthly income between 10,001–20,000 INRs (29%), had moderate depressive symptom severity at baseline (PHQ-9 score of 13), an HbA1c level of 8.9% at baseline, and indicated diabetes as their most stigmatizing condition (84%).

### Longitudinal analysis of self-stigma scores

At baseline, while holding all other covariates constant, CoCM participants had an average SSCI-4 score that was 0.32 points lower than enhanced standard of care participants (95%CI: -0.56, -0.079; $P < .01$). Over the 12-month follow-up period, while holding all other covariates constant, SSCI-4 scores decreased on average by 0.011 points in total for enhanced standard-of-care participants (95% CI: -0.019, -0.0033; $P < .01$). Among CoCM participants, while holding all other covariates constant, the mean rate of monthly change in SSCI-4 scores was 0.0087 points higher than for the enhanced standard of care group (95% CI: -0.0018, 0.019; $P = .10$). Thus after 12 months, a CoCM participant with mean covariate predictor values would have a predicted SSCI-4 score of 4.5 on average (a decrease of 0.028 relative to baseline) compared to 4.7 on average for the enhanced standard of care group (a decrease of 0.13 relative to baseline). See Table 2 for full results.

**Table 1. Descriptive characteristics of study sample.**

| | Enhanced standard-of-care (control, n = 208) | CoCM (intervention, n = 195) | All participants (n = 403) |
|---|---|---|---|
| **Sex** | | | |
| Male | 76 (37%) | 89 (46%) | 165 (41%) |
| Female | 132 (64%) | 106 (54%) | 239 (59%) |
| **Age** | | | |
| 30–39 years | 14 (6.7%) | 15 (7.7%) | 29 (7.2%) |
| 40–49 years | 56 (27%) | 58 (30%) | 114 (28%) |
| 50–59 years | 82 (39%) | 79 (40%) | 161 (40%) |
| 60 years or older | 56 (27%) | 43 (22%) | 99 (25%) |
| **Level of educational attainment** | | | |
| Less than primary school | 20 (9.6%) | 18 (9.2%) | 38 (9.4%) |
| Primary school | 30 (14%) | 35 (18%) | 65 (16%) |
| Secondary school | 108 (52%) | 99 (51%) | 207 (51%) |
| Post-secondary school | 48 (23%) | 43 (22%) | 91 (23%) |
| Missing | 2 (1.0%) | 0 (0%) | 2 (0.5%) |
| **Average monthly household income (INRs)** | | | |
| <3,000 | 4 (1.9%) | 6 (3.1%) | 10 (2.5%) |
| 3,000–10,000 | 62 (30%) | 48 (25%) | 110 (27%) |
| 10,001–20,000 | 60 (29%) | 57 (29%) | 117 (29%) |
| 20,001–30,000 | 39 (19%) | 36 (18%) | 75 (19%) |
| 30,001–40,000 | 12 (5.8%) | 12 (6.1%) | 24 (5.9%) |
| 40,001–50,000 | 11 (5.3%) | 13 (6.6%) | 24 (5.9%) |
| >50,000 | 20 (9.6%) | 23 (12%) | 43 (11%) |
| **Self-reported stigmatized condition** | | | |
| Depression | 27 (13%) | 17 (9%) | 44 (11%) |
| Diabetes | 173 (83%) | 167 (86%) | 340 (84%) |
| Missing | 8 (3.8%) | 11 (5.6%) | 19 (4.7%) |
| **Self-stigma score at baseline (SSCI; 4–20)** | | | |
| Median [IQR] | 4 [4–5] | 4 [4,4] | 4 [4,4] |
| Min, Max | 4, 17 | 4, 15 | 4, 17 |
| Missing | 5 (2.4%) | 3 (1.5%) | 8 (2.0%) |
| **Change in self-stigma score (SSCI; range: -16, +16)** | | | |
| Median [IQR] | 0 [0–0] | 0 [0–0] | 0 [0–0] |
| Min, Max | -13, 7 | -10, 5 | -13, 7 |
| Missing | 12 (5.8%) | 14 (7.1%) | 26 (6.4%) |
| **Depression at baseline (PHQ9; range: 0–27)** | | | |
| Median [IQR] | 13 [12–15] | 13 [11–14] | 13 [11–14] |
| Min, Max | 10, 21 | 10, 22 | 10, 22 |
| **Depression at endline (PHQ9; range: 0–27)** | | | |
| Median [IQR] | 7 [3–11] | 4 [3–7] | 5 [3–9] |
| Min, Max | 0, 22 | 0, 21 | 0, 22 |
| Missing | 6 (2.9%) | 4 (2.1%) | 10 (2.5%) |
| **Diabetes at baseline (A1c)** | | | |
| Median [IQR] | 8.7 [7.7–10] | 9.1 [7.8–11] | 8.9 [7.7–11] |
| Min, Max | 5.6, 15.0 | 5.8, 15.2 | 5.6, 15.2 |
| **Diabetes at endline (A1c)** | | | |
| Median [IQR] | 8.2 [7.1–9.4] | 7.8 [6.7–9.3] | 8.0 [6.9–9.3] |
| Min, Max | 4.4, 14.3 | 5.3, 13.9 | 0, 14.3 |
| Missing | 6 (2.9%) | 4 (2.1%) | 10 (2.5%) |

**Table 2. Predictors of self-stigma scores longitudinally.**

| Predictors | Point estimates | 95% CI | P-value |
|---|---|---|---|
| Intercept | 5.3 | 4.4, 6.3 | < .001 |
| Collaborative Care Model | -0.32 | -0.56, -0.079 | < .01 |
| Time (12 months) | -0.011 | -0.019, -0.0039 | < .01 |
| Collaborative Care Model*Time | 0.0087 | -0.0018, 0.019 | .10 |
| Age | -0.0031 | -0.012, 0.0062 | .52 |
| Sex (Female) | 0.00047 | -0.16, 0.16 | 1.0 |
| Monthly household income (INRs) | | | |
| 3,000–10,000 | -0.46 | -0.99, 0.066 | .086 |
| 10,001–20,000 | -0.56 | -1.1, -0.041 | < .05 |
| 20,001–30,000 | -0.56 | -1.1, -0.031 | < .05 |
| 30,001–40,000 | -0.55 | -1.2, 0.055 | .075 |
| 40,001–50,000 | -0.35 | -0.96, 0.25 | .25 |
| >50,000 | -0.40 | -0.96, 0.17 | .17 |
| Level of educational attainment | | | |
| Primary school | 0.028 | -0.28, 0.34 | .86 |
| Secondary school | -0.34 | -0.61, -0.069 | < .05 |
| Post-secondary school | -0.43 | -0.75, -0.11 | < .01 |
| PHQ-9 scores at baseline | -0.026 | -0.058, 0.0068 | .12 |
| HbA1c levels at baseline | 0.092 | 0.050, 0.13 | < .001 |

## Mediation analysis

In the unadjusted analysis (see Table 3), for participants with a 1-unit drop in self-stigma scores over 12 months, HbA1c levels were on average 0.070% lower (total effect; 95% CI: 0.0032, 0.14, $P < .05$). However, we found no statistically significant differences when accounting for depressive symptoms as a mediator (direct effect; B = 0.064, 95% CI: -0.0043, 0.13, $P = .069$) and the difference in the total effect minus the direct effect (average causal mediation effect; B = 0.0062, 95% CI: -0.0023, 0.019, $P = .16$). Thus, this direct effect estimate confirms that a change in self-stigma did not have an effect on diabetes outcomes through depressive symptoms as a mediator.

We also conducted a sensitivity analysis to adjust for the covariates of age, sex, monthly household income, and level of educational attainment (see Table 4). In the adjusted analysis, while holding all other covariates constant, for participants with a 1-unit drop in self-stigma scores over 12 months, HbA1c levels were on average 0.071% lower (total effect; 95% CI: 0.0039, 0.14, $P < .05$). Like the unadjusted analysis though, we found no statistically significant differences when accounting for depressive symptoms as a mediator, (direct effect; B = 0.065, 95% CI: -0.0040, 0.13, $P = .065$) and the difference in the total effect minus the direct effect

**Table 3. Unadjusted total effect, direct effect, average causal mediated effect, and proportion mediated.**

| Effects on diabetes after 12 months | Point estimates | 95% CI | P-value[1] |
|---|---|---|---|
| Average causal mediated effect | 0.0062 | -0.0023, 0.019 | .16 |
| Average direct effect | 0.064 | -0.0043, 0.13 | .069 |
| Total effect | 0.070 | 0.0032, 0.14 | < .05 |
| Proportion mediated | 0.090 | -0.093, 0.64 | .20 |

[1]All P-values calculated using bootstrapping.

**Table 4. Adjusted total effect, direct effect, average causal mediated effect, and proportion mediation.**

| Adjusted effects on diabetes after 12 months[1] | Point estimates | 95% CI | P-value[2] |
|---|---|---|---|
| Average causal mediated effect | 0.0062 | -0.0011, 0.019 | .12 |
| Average direct effect | 0.065 | -0.0040, 0.13 | .065 |
| Total effect | 0.071 | 0.0039, 0.14 | < .05 |
| Proportion mediated | 0.089 | -0.044, 0.69 | .20 |

[1]Effects adjusted for age, sex, monthly household income, and level of educational attainment.

[2]All P-values calculated using bootstrapping.

(average causal mediation effect; B = 0.0062, 95% CI: -0.0011, 0.019, P = .12). Thus, this estimate confirms that a change in self-stigma did not have an effect on diabetes outcomes through depressive symptoms as a mediator even after adjusting for co-variates.

## Discussion

Compared to the enhanced standard-of-care participants, self-stigma for CoCM participants increased marginally over time, contrary to what we anticipated. However, this relationship was not statistically significant nor was the magnitude of change in self-stigma scores longitudinally–among participants who on average at baseline indicated experiencing almost no self-stigma–large enough to demonstrate any noticeable clinical change. We found that there was no total effect of change in self-stigma over 12 months on diabetes outcomes at 12 months and that depressive symptom severity at 12 months was not a mediator.

We grounded our choice of using the change in the total self-stigma score as the predictor because previous studies show that people living with common mental disorders experienced stigma from providers in India [22,23,25]. We hypothesized that CoCM would be a 'stigma-free' way of treating depression in a diabetes context, thereby suggesting that treated depression would mediate the relationship between change in self-stigma and diabetes outcomes. However, given our findings, an alternative analytical approach such as structural equations modeling could investigate the direction of association.

Our study results contribute to a rich and expanding literature at the intersection of stigma, common mental disorders, and diabetes in India by adding, to our knowledge, the first empirical study examining self-stigma in the context of an integrated BH intervention for people with comorbid depression and type 2 diabetes. Other interventions addressing stigma among people living with mental illness in India have typically focused on raising community awareness or knowledge about mental illness with the goal of reducing stigmatizing behaviors and none have specifically targeted healthcare professionals [23]. Armstrong and colleagues observed modest reductions in stigmatizing attitudes towards people with common mental disorders among allied community health workers serving in Bengaluru Rural District in Karnataka, following a health literacy training intervention [51]. Similarly, Mindlis and colleagues observed lower stigma levels following an educational intervention targeting six villages in rural Vadodara District in Gujarat [52]. In the Systematic Medical Appraisal Referral and Treatment Mental Health Project in West Godavari District of Andhra Pradesh, researchers used a public, multi-component anti-stigma campaign and observed a score improvement in stigma perceptions related to healthcare seeking [53,54]. We are unaware of any interventions aimed at reducing diabetes-related stigma in India and diabetes-related stigma interventions globally are very understudied with a recent systematic review identifying no stigma-reducing interventions in healthcare facilities for diabetes [55].These findings highlight the importance

of a multi-pronged approach to reducing stigma that incorporates community-level prevention efforts and programs that focus on healthcare providers and systems [25].

Although some studies suggest people living with diabetes in India face stigma [18–21], we assumed in our proposed pathway that depression would be the most stigmatizing condition. Yet 340 participants (84%) indicated that diabetes was their most stigmatized condition. During the formative evaluation, healthcare providers indicated that patients would be reluctant to disclose their mental illness, which could underscore the high percentage of participants choosing diabetes [56]. Research staff members anecdotally observed in this study that participants may have reflexively selected diabetes given that they were at a diabetes clinic during the study procedures or because it is a more visible condition. During the process evaluation of implementing CoCM during the INDEPENDENT trial, some participants expressed that their depression was a symptom of their diabetes or that they may not have believed they had a mental disorder [57]. This may in turn explain why so few participants self-reported depression as their most stigmatizing condition.

Whereas the four questions contained in the SSCI-4–avoidance, blame, feeling left out, and embarrassment–are central to self-stigma in the context of mental illness in settings in the Global North [1,36], their transferability to settings in India warrants further investigation. In a multi-site study in South India among people with schizophrenia, Koschorke and colleagues found patients experienced internalized stigma more frequently than they experienced negative discrimination [58]. In contrast, Goyal and colleagues observed low levels of internalized stigma in a cross-sectional study of patients with psychiatric disorders in India [59]. In their qualitative thematic analysis exploring community level notions of stigma in Kerala, Raghavan and colleagues found low levels of self-stigma and while the construct of self-stigma could partially explain the lived experiences of people with mental illness, there were other locally relevant notions [60]. These included the impairment of marriage prospects as a result of mental illness; the duality of family, friends, and/or community members as supporters and/or as antagonists; and the role of the collective (family, community, or other groups) in producing and upholding stigma [60]. In India, these collective notions of stigma intersect with gender, caste, and religion (among other factors). Using the SSCI-4 as an instrument to measure self-stigma might be less relevant than other scales for measuring societal stigma. As such, our research underscores the need for culturally grounded, valid, and reliable instruments to capture multi-dimensional stigma.

## Limitations

Our study has several important limitations that warrant consideration. The researchers in the primary trial were unable to collect caste data, which represents a missed opportunity to examine intersectional stigma. Due to practical constraints, we were unable to locally adapt and validate the SSCI-4 at each study site and considering the four different sites are heterogenous with respect to language, religion, and other socio-demographic factors, the scale may not have been sensitive to measuring self-stigma at the different sites. Furthermore, SSCI-4 scores were collected only at yearly intervals in this longitudinal study (unlike PHQ-9 and HbA1c scores which were collected every six months), which hindered the ability to examine predictor, mediator, and outcome scores with more temporal sensitivity. Lastly, participants with moderate to severe depressive symptom severity were recruited and in the context of the risk of regression to the mean [61,62], our analysis–which looks at depressive symptom severity at 12 months cross-sectionally–might be looking at attenuated effects.

There are three potential sources of bias in mediation analyses like this which can increase the risk of incorrect interpretations and inference. These include collider stratification bias

(mediator-outcome confounding), mediator-outcome confounding affected by the predictor, and exposure-mediator interactions [63]. Collider stratification bias is particularly relevant if there are unknown or unmeasured confounders on the mediator and outcome, thus inducing a spurious effect between the mediator and predictor [63,64]. While the trial design balanced the distribution of known confounders at baseline to reduce selection bias, there remains a risk of an unmeasured confounder with respect to change in self-stigma as a predictor in this secondary mediation analysis.

## Conclusion

Self-stigma scores longitudinally did not significantly differ when comparing participants who received CoCM to those who received enhanced standard-of-care. Our study findings show that the effect of change in self-stigma on diabetes outcomes are not mediated by depressive symptoms in the context of CoCM implementation at four urban diabetes clinics in India. However, given the narrow scope of our study and the local and plural notions of stigma in India, we recommend further research on the use of culturally grounded instruments to assess self-stigma. While integrated care delivery models like CoCM hold promise to improve health outcomes, our findings suggest that embedding additional and multi-level stigma-reduction interventions, including those operating at the community and healthcare systems levels, may be needed to reduce self-stigma.

## Supporting information

**S1 Checklist. Inclusivity in global research.**
(DOCX)

**S1 File. 4-item self-stigma scale for chronic Illness.**
(DOCX)

**S2 File. Annotated statistical software code.**
(DOCX)

## Acknowledgments

This publication is adapted from a chapter from a published doctoral dissertation (Halliday SK. Collaborative Care Models for Behavioral Healthcare Services Integration: ProQuest Dissertations Publishing; 2023). We acknowledge valuable input from Dr. Julia Dombrowski on refining the conceptual pathway model and analytical approach. We would like to give our sincerest thanks to all the participants in the study. Lastly, we are grateful to all the healthcare staff members at the study sites whose dedication and service continually inspire us.

## Author Contributions

**Conceptualization:** Scott Halliday, Deepa Rao, Orvalho Augusto, Subramani Poongothai, Gumpeny R. Sridhar, Nikhil Tandon, K. M. Venkat Narayan, Leslie C. M. Johnson, Bradley H. Wagenaar, David Huh, Brian P. Flaherty, Lydia A. Chwastiak, Mohammed K. Ali, Viswanathan Mohan.

**Data curation:** Aravind Sosale, Rajesh Sagar, Shivani A. Patel.

**Formal analysis:** Scott Halliday, Orvalho Augusto.

**Funding acquisition:** Deepa Rao, K. M. Venkat Narayan, Mohammed K. Ali.

**Investigation:** Scott Halliday, Orvalho Augusto.

**Methodology:** Scott Halliday, Deepa Rao, Orvalho Augusto, Bradley H. Wagenaar, David Huh, Brian P. Flaherty.

**Project administration:** Scott Halliday, Deepa Rao, Mohammed K. Ali.

**Resources:** Deepa Rao, Mohammed K. Ali.

**Software:** Scott Halliday, Orvalho Augusto.

**Supervision:** Deepa Rao, Mohammed K. Ali.

**Validation:** Scott Halliday, Orvalho Augusto.

**Visualization:** Scott Halliday.

**Writing – original draft:** Scott Halliday.

**Writing – review & editing:** Scott Halliday, Deepa Rao, Orvalho Augusto, Subramani Poongothai, Aravind Sosale, Gumpeny R. Sridhar, Nikhil Tandon, Rajesh Sagar, Shivani A. Patel, K. M. Venkat Narayan, Leslie C. M. Johnson, Bradley H. Wagenaar, David Huh, Brian P. Flaherty, Lydia A. Chwastiak, Mohammed K. Ali, Viswanathan Mohan.

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
