## [Decision Letter · Decision Letter 0]

27 Jun 2024

PGPH-D-24-00397

A mediation analysis evaluating change in self-stigma on diabetes outcomes among people with depression in urban India: A secondary analysis from the INDEPENDENT trial of the Collaborative Care Model

Dear Dr. Halliday,

Thank you for submitting your manuscript to PLOS Global Public Health. After careful consideration, we feel that it has merit but does not fully meet PLOS Global Public Health’s publication criteria as it currently stands. Therefore, we invite you to submit a revised version of the manuscript that addresses the points raised during the review process.

We look forward to receiving your revised manuscript.

Kind regards,

Parvati Singh, PhD

Academic Editor

Journal Requirements:

Reviewers' comments:

Reviewer's Responses to Questions

**Comments to the Author**

1. Does this manuscript meet PLOS Global Public Health’s publication criteria? Is the manuscript technically sound, and do the data support the conclusions? The manuscript must describe methodologically and ethically rigorous research with conclusions that are appropriately drawn based on the data presented.

Reviewer #1: Yes

2. Has the statistical analysis been performed appropriately and rigorously?

Reviewer #1: I don't know

3. Have the authors made all data underlying the findings in their manuscript fully available (please refer to the Data Availability Statement at the start of the manuscript PDF file)?

Reviewer #1: Yes

4. Is the manuscript presented in an intelligible fashion and written in standard English?

Reviewer #1: Yes

5. Review Comments to the Author

Reviewer #1: The current manuscript described the relationship between changes in self-stigma and diabetes outcomes in an Indian population with diabetes and depression. Stigmatization is increasingly considered a relevant and important topic in the field of diabetes. There is a modest amount of studies on stigmatization in people with diabetes, and additional studies are needed and welcome. Cultural differences in (diabetes) stigma are known, but hardly studied and described. Therefore, this study adds to existing knowledge and study results are relevant to the field of diabetes and clinical practice.

The current study performed a mediation-analysis to examine whether depression would mediate the relationship between changes in self-stigma and HbA1c. Decreases in self-stigma scores predicted HbA1c outcomes, but were not mediated by depression scores.

The current manuscript is well written and structured and based upon relevant literature. Potential mechanisms that may explain study results are described in the discussion section.

Methodology and statistical analyses appear sound and additional information on the mediation-analyses is provided.

Some considerations are put forth, for the authors here:

Introduction:

-p3, line 57-58 ‘’Importantly, negative attitudes and beliefs associated with self-stigma can be reinforced by healthcare providers..’’ It is presented as something commonly known, but I believe there are strong cultural differences in how healthcare providers could reinforce self-stigma. Could you further explain what you mean by this, in your Indian context?

-p3, line 62-63 ‘’Visiting a specialist mental health care provider….mental disorders’.’ Could you further explain the mechanism behind or link between visiting mental healthcare providers and stereotyping, prejudice and discrimination?

-In the introduction it is not clear what type of diabetes participants were diagnosed with. Is it all type 2 diabetes? Or type 1 diabetes as well?

Methods:

-p4, line 95-100 ‘’In this pragmatic trial ….. 36 months’’. This is a very long and difficult to read sentence. Could you change it into several shorter sentences?

-p4, line 100 ‘’To be eligible, participants must have been 35 years…’’ Why 35? Could you explain why younger adults would not be included? -> see also my comment on type of diabetes (introduction), that might be related to this.

-p.7, Figure 1: Why did the authors chose to compare changes in self-stigma to HbA1c at 12 months? Why not look at the relationship between change in self-stigma and change in HbA1c and change in depression? I am no statistician, but conceptually I would think that it makes sense to compare changes in all 3 variables. Especially since there seems to be quite a change in depression scores and HbA1c from baseline to 12 months.

Results:

-p8, line 173: ‘’and indicated diabetes as their most stigmatizing condition’’ = double, please remove.

6. PLOS authors have the option to publish the peer review history of their article (what does this mean?). If published, this will include your full peer review and any attached files.

**Do you want your identity to be public for this peer review?** For information about this choice, including consent withdrawal, please see our Privacy Policy.

Reviewer #1: No

---

## [Editor Report · Decision Letter 1]

1 Aug 2024

A mediation analysis evaluating change in self-stigma on diabetes outcomes among people with depression in urban India: A secondary analysis from the INDEPENDENT trial of the Collaborative Care Model

PGPH-D-24-00397R1

Dear Dr. Halliday,

We are pleased to inform you that your manuscript 'A mediation analysis evaluating change in self-stigma on diabetes outcomes among people with depression in urban India: A secondary analysis from the INDEPENDENT trial of the Collaborative Care Model' has been provisionally accepted for publication in PLOS Global Public Health.

Best regards,

Parvati Singh, PhD

Academic Editor